# Additive manufacturing of alloys with programmable microstructure and properties

Shubo Gao [1,2], Zhi Li [3], Steven Van Petegem [4], Junyu Ge[1], Sneha Goel [4,5,9], Joseph Vimal Vas[6], Vladimir Luzin[7], Zhiheng Hu [2], Hang Li Seet[2], Dario Ferreira Sanchez [4], Helena Van Swygenhoven[4], Huajian Gao [1,3] & Matteo Seita [8] ✉

In metallurgy, mechanical deformation is essential to engineer the microstructure of metals and to tailor their mechanical properties. However, this practice is inapplicable to near-net-shape metal parts produced by additive manufacturing (AM), since it would irremediably compromise their carefully designed geometries. In this work, we show how to circumvent this limitation by controlling the dislocation density and thermal stability of a steel alloy produced by laser powder bed fusion (LPBF) technology. We show that by manipulating the alloy's solidification structure, we can 'program' recrystallization upon heat treatment without using mechanical deformation. When employed site-specifically, our strategy enables designing and creating complex microstructure architectures that combine recrystallized and non-recrystallized regions with different microstructural features and properties. We show how this heterogeneity may be conducive to materials with superior performance compared to those with monolithic microstructure. Our work inspires the design of high-performance metal parts with artificially engineered microstructures by AM.

Many metal manufacturing technologies used in our society rely on a combination of mechanical and thermal processes to shape the material into the desired geometry and concurrently engineer its microstructure and properties[1-3]. The amount of mechanical strain introduced during forging and extrusion of metals, for instance, can be controlled to harden the material via dislocation accumulation, or trigger microstructure recrystallization upon heat treatment (HT)—a phenomenon by which newly defect-free grains nucleate and grow, yielding higher toughness and more isotropic mechanical properties.

This archetypical "heat and beat" approach—which is in use since the bronze age—breaks down when employing modern additive manufacturing (AM) technologies. AM, also known as three-dimensional (3D) printing, enables joining materials together layer by layer to produce near-net-shape parts with previously impossible geometric complexity[4,5]. Because material and geometry are formed simultaneously during AM, further mechanical processing of the solid to drive controlled microstructural changes is not possible without irremediably compromising the part's intricate shape[6]. Thus,

[1]School of Mechanical and Aerospace Engineering, Nanyang Technological University, Singapore 639798, Republic of Singapore. [2]Additive Manufacturing Division, Singapore Institute of Manufacturing Technology (SIMTech), Agency for Science, Technology and Research (A*STAR), Singapore 636732, Republic of Singapore. [3]Institute of High Performance Computing, Agency for Science, Technology and Research (A*STAR), Singapore 138632, Republic of Singapore. [4]Photon Science Division, Paul Scherrer Institute, Villigen 5232, Switzerland. [5]VTT Technical Research Centre of Finland, Espoo 02150, Finland. [6]School of Materials Science and Engineering, Nanyang Technological University, Singapore 639798, Republic of Singapore. [7]Australian Nuclear Science & Technology Organisation (ANSTO), Lucas Heights, NSW 2234, Australia. [8]Department of Engineering, University of Cambridge, Cambridge CB2 1PZ, UK. [9]Present address: Advanced materials for nuclear energy, VTT Technical Research Centre of Finland, Espoo 02150, Finland. ✉e-mail: ms2932@eng.cam.ac.uk

compared to traditional manufacturing routes, AM offers less opportunities to control the microstructure of metals and tailor their properties. For this reason, intense research is focused on devising AM processes aimed at optimizing the as-printed microstructure.

Here, we show how to control the microstructure evolution of additively manufactured stainless steel without relying on mechanical deformation. Using laser powder bed fusion (LPBF) technology, we devise processing strategies to 'program' the thermal stability of the as-printed alloy, such that it is possible to decide, a priori, how the material's microstructure will evolve upon HT. These strategies restore some of the microstructure control capabilities offered by conventional metal processing. More importantly, they allow creating new materials by programming the alloy's microstructure site-specifically, in 3D, and at high spatial resolution. Site-specific microstructure control of metals is one of the most unique and attractive capabilities of AM[7–9]. Our strategies showcase an advanced approach to this paradigm by enabling direct control over the evolution of multiple microstructural features at once, broadening the design space of engineering materials with optimized mechanical and physical properties.

## Results

### Programmable and site-specific recrystallization

In conventional thermo-mechanical processes, plastic deformation of metals induces dislocation accumulation in the polycrystalline microstructure, which hardens the metal and makes it less malleable. Upon HT, the strain energy stored in dislocations drives microstructural changes that restore the metal's original plasticity. Heat promotes dislocation annihilation−a process referred to as recovery−and may trigger recrystallization[10]. During recrystallization, new grains grow into the dislocated crystals, yielding a refined and equiaxed microstructure. It is through carefully designed combinations of mechanical and thermal treatments that the microstructure evolution of metals can be engineered to fine-tune their properties.

In our previous work[11], we confirmed that recrystallization of stainless steel 316L (SS316L) produced by LPBF is driven by similar mechanisms to those at play in conventionally produced metals, in that it requires a minimum, critical, dislocation density (i.e., strain energy) to be triggered at a certain temperature. This is typically controlled through mechanical deformation. We also found that the propensity of the material to complete recrystallization depends on the extent of chemical heterogeneity found in the as-solidified microstructure. Pronounced micro-segregation of solute across the microstructure hinders grain boundary motion and yields materials that retain most of their as-solidified microstructural characteristics, even after HT. Conversely, microstructures with more homogeneous chemical composition exhibit lower thermal stability and hence higher propensity to undergo complete recrystallization upon thermo-mechanical processing.

Thus, gaining control over the dislocation density and the alloy's chemical heterogeneity directly during LPBF would allow engineering the microstructure evolution of the material upon HT and drive recrystallization 'on demand', without the need for any additional mechanical deformation. To achieve this, we devise two distinct LPBF processing strategies denoted as "H" and "L" (detailed in "Methods") to produce SS316L with high and low thermal stability, respectively. During LPBF, a thin layer of metal powders is spread over a build platform and then selectively melted by a high-power laser source (Fig. 1a). Melting occurs sequentially as the laser is scanned back and forth in tracks that fill the areas to be consolidated. The H and L strategies differ in how close melt tracks are to one another within a single layer (i.e., the melt track area density), which is determined by the hatch spacing parameter, $h$. They also differ in how many times the same layer is scanned by the laser; namely whether the previously consolidated material is remelted or not. As we detail in the following section, by varying $h$ we control the density of geometrically necessary dislocations (GNDs)−which accommodate local plastic strains gradients−while by remelting we promote homogenization of the local chemical composition and thus attenuate micro-segregation.

After a HT at 1050 °C for 30 min, SS316L samples produced using strategy H (single laser scanning with a large $h$ value, H-SS316L) exhibit a microstructure similar to the as-printed state (which is reported in Supplementary Fig. 1a). As shown in Fig. 1b, this microstructure consists of large columnar grains sharing a common <110> crystallographic texture aligned parallel to the build direction, localized crystal misorientation spreads, and a high fraction of low-angle grain boundaries. The area-weighted average grain size in this microstructure is 120 μm. By contrast, SS316L samples produced using strategy L (laser remelting using a small $h$ value, L-SS316L) undergo complete recrystallization. Through recrystallization, the microstructure changes substantially in comparison to the as-printed L-SS316L state (which is reported in Supplementary Fig. 1b). As shown in Fig. 1c, it consists of randomly oriented grains with equiaxed morphology and an average size of 20 μm, which are GND-free and separated by a wide range of different grain boundaries, including many twin boundaries.

To demonstrate that recrystallization can be programmed site-specifically, we encode two lines of binary code corresponding to the letters "A" and "M" (the acronym of additive manufacturing) in the microstructure of a single block of SS316L (Fig. 1d). Noteworthy is that this level of control over the microstructure is three-dimensional. Recrystallization may be controlled both in-plane, within each layer (as it is the case in Fig. 1d), as well as out-of-plane in a layered configuration along the build direction (as shown in Fig. 1e). These results demonstrate that our thermal stability control enables complete design freedom to 'architect' the microstructure of SS316L with different fractions and distributions of each microstructure constituent. These LPBF strategies are also machine-agnostic. In fact, we produced the two samples shown in Fig. 1d, e using different LPBF machines. We highlight, however, that the exact laser parameters employed for strategy H and L differ when using different machines (as we detail in "Methods") and should be revised depending on the size and geometry of the build. The smaller the laser-scanned areas to be consolidated, the higher the retained heat during processing, which may affect the solidification microstructure significantly.

To the best of our knowledge, the 3D microstructure design capabilities achieved in this work are unmatched in the state of the art. Specially designed thermo-mechanical processes involving accumulative roll bonding[12] and asymmetric rolling[13] have only been successful at creating two-dimensional architectures that combine lamellae of recrystallized and non-recrystallized microstructures. By contrast, our LPBF strategies may be alternated in space at a relatively high spatial resolution (as shown in Fig. 1e) and offer an entirely deformation-free, site-specific control over the microstructure, which has not been achieved so far. Moreover, they could be used to site-specifically modify multiple microstructural features at once, including crystallographic texture, grain morphology, dislocation density, and grain boundaries character distribution (as we show in Fig. 1 and Supplementary Fig. 2).

### Control of microstructural features governing thermal stability

LPBF materials can essentially be thought of as stacks of overlapping melt pools (Fig. 2a), each one producing a localized thermo-mechanical treatment. The rapid heating and cooling associated with each melt pool, in fact, induces expansion and contraction of the material, generating plastic strains and thus driving dislocation accumulation[14,15]. It follows that the total plastic strain in the as-printed material depends on the number of melt pools per unit area per layer, or the number of melt tracks per unit length. This number may be directly controlled by the hatch spacing, $h$. For a fixed melt pool width,

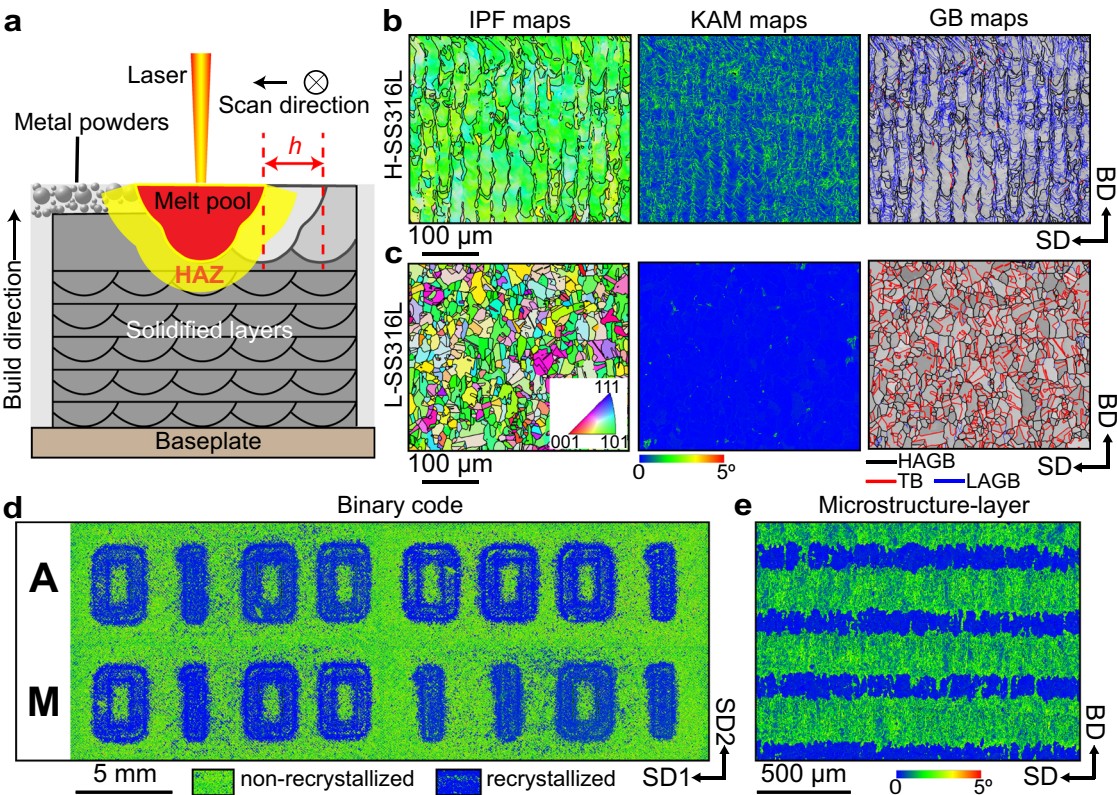

**Fig. 1 | Programmable thermal stability in stainless steel 316L (SS316L) by laser powder bed fusion (LPBF). a** Schematic cross-section view of a metal sample produced by LPBF, which shows the melt pool and the resulting heat affected zone (HAZ). **b**, **c** Electron backscatter diffraction maps showing crystal orientation (IPF, inverse pole figure) along the build direction (BD), kernel average misorientation (KAM), and grain boundary (GB) character distribution in heat-treated H-SS316L

and L-SS316L, respectively. HAGB, LAGB, and TB represent high-angle grain boundaries, low-angle grain boundaries, and twin boundaries, respectively. SD represents scanning direction. **d**, **e** KAM maps illustrating the site-specific control over recrystallization both within the build plane and along the BD, respectively. The binary code in (**d**) stands for "AM".

$w$, a smaller value of $h$ leads to the occurrence of more thermal cycles per unit length, producing larger cumulative plastic strains in the material and thus larger dislocation densities. We test this hypothesis using a combination of experiments and simulations. We calculate the GND density based on the average local misorientation angle measured by EBSD (see "Methods") and estimate the plastic strain in the material by setting up a finite element model (FEM) of the LPBF process in Abaqus (detailed in "Methods"). We show representative FEM images of the cumulative plastic strain distribution in the builds in Supplementary Fig. 3. We plot results from measurements and simulations side by side in Fig. 2b and as a function of the ratio $h/w$. The experimental melt pool width $w$ (70 μm here) was determined by averaging the width of over 20 melt pools from optical micrographs of etched cross-sections (Fig. 2a). To be able to compare the two sets of results directly, we link the GND density to the simulated plastic strain using the formula proposed by Ashby[16], which has been successfully applied to alloys produced by LPBF[17]:

$$\rho_{GND} = \frac{\varepsilon}{4b\lambda}. \tag{1}$$

Here, $\varepsilon$ is the simulated plastic strain, $b$ is the magnitude of the Burgers vector in SS316L (0.25 nm), and $\lambda$ is the average size of the solidification cells in the as-printed samples (~308 nm). The data shows good agreement between measurements and simulations, confirming our hypothesis that varying $h$ allows tuning the GND density in the material—and thus the driving force for recrystallization—without the need for additional mechanical deformation.

It is interesting to note that the simulations consistently underestimate the measurements. We ascribe this discrepancy to the fact that the FEM model does not capture the non-uniform distribution of plastic strain in each melt pool, which stems from the underlying solidification microstructure. As we show in the KAM maps from the top and side view of our samples (Fig. 2c), crystal misorientation (and thus GND density) is highest at melt pool centerlines. This phenomenon has been imputed to the coalescence of differently oriented cell structures, which grow from the melt pool side walls inward to meet at the centerline[18,19]. Here, local misorientations are accommodated by GNDs. As melt pools overlap during LPBF, a fraction of the GND density is lost through remelting and annealing of the material. In FEM simulations, this loss may be overestimated because the plastic strain distribution is assumed uniform within the melt pool volume, whereas in reality the highest density of GNDs is concentrated at centerlines (which are not remelted).

From our investigations (Fig. 2d), the microstructure produced using $h = 10$ μm ($h/w = 0.14$) contains 90% higher GND density compared to that produced using $h = 35$ μm ($h/w = 0.5$). Upon HT, the latter shows almost no recrystallization, and indeed remains stable up to 1200 °C (see Supplementary Fig. 4a). Conversely, the former exhibits a partially recrystallized microstructure (see Supplementary Fig. 4b). The reason why this microstructure undergoes only partial recrystallization despite having the highest GND density in our dataset is that recrystallized grains are pinned by a more pronounced microsegregation at cell boundaries.

During solidification, solute is rejected from the solid phase into the liquid phase to form a thin solute-enriched film that follows the solidification front as it moves[20]. Under rapid solidification conditions,

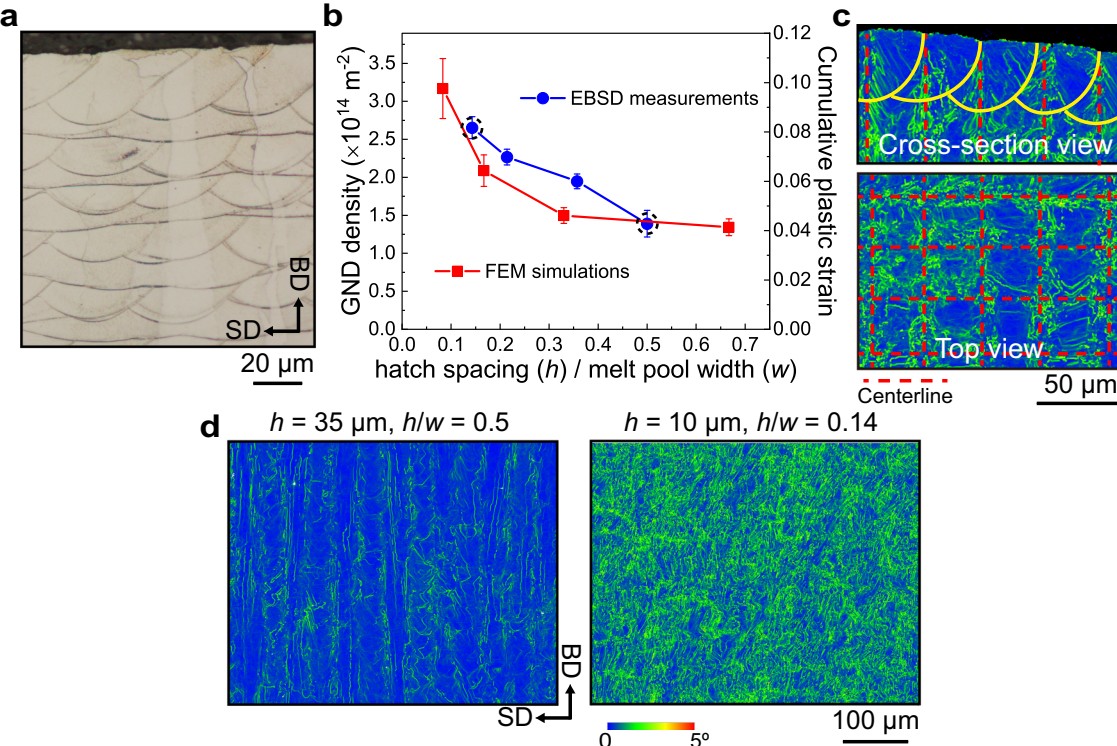

**Fig. 2 | Tailoring the driving force for recrystallization by varying laser hatch spacing. a** Optical micrograph showing the arrangement of melt pools in our samples. **b** Estimated geometrically necessary dislocation (GND) densities as measured by EBSD and from the cumulative plastic strains obtained through finite element model (FEM). The values refer to as-built SS316L samples. The experimental melt pool width ($w$) is 70 μm in this work. **c** Cross-section EBSD KAM maps from the top surface and top view KAM map from the center region of an SS316L sample produced using a hatch spacing of 35 μm ($h$ = 35 μm). Yellow curves indicate melt pool boundaries. Red dashed lines indicate centerlines. **d** KAM maps highlighting the difference in crystal misorientation distribution between samples produced using $h$ = 35 μm and $h$ = 10 μm, which correspond to the first and last data point (enclosed within a dashed circle) in (**b**), respectively.

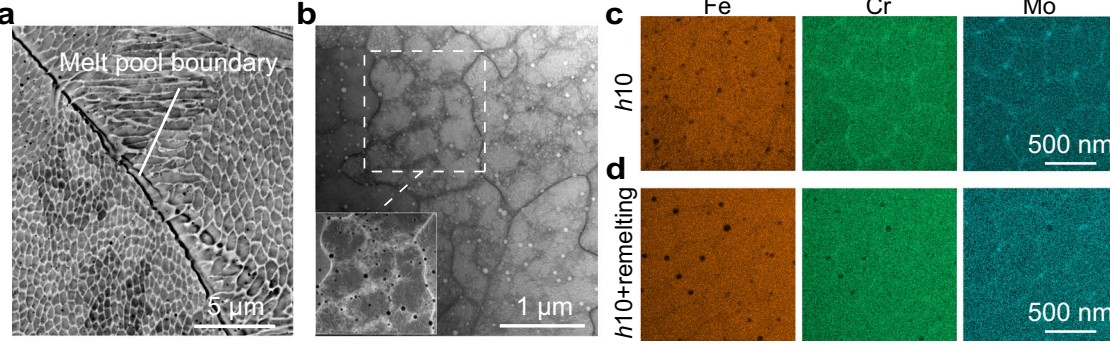

**Fig. 3 | Manipulating the solidification structure by remelting. a** A scanning electron micrograph of chemically etched SS316L showing the morphology of the solidification cells intersecting the sample surface. Solute-enriched cell boundaries protrude from the surface (and thus appear bright in the micrograph) as they are more resistant to chemical attack compared to the matrix. **b** Scanning transmission electron microscopy (STEM) analysis of the internal structure of cell boundaries, which exhibits high dislocation density. **c, d** Energy disperse spectroscopy (EDS) STEM measurements comparing solute distribution between two $h$10 samples (using a hatch spacing of 10 μm) produced without and with laser remelting, respectively.

which are typical in LPBF, this film remains kinetically trapped at cell boundaries, creating a microscopic chemical heterogeneity in the alloy (Fig. 3a). This micro-segregation acts as a barrier to grain boundary migration[21] and thus hinders recrystallization[11]. The higher the density of cells and the higher the amount of solute segregating at their boundaries, the higher the alloy's thermal stability.

Through scanning transmission electron microscopy (STEM) analysis of the sample produced using $h$ = 10 μm (shown in Fig. 3b, c), we find that cell boundaries are decorated with copious dislocations and a higher concentration of chromium (Cr, 19.8%, wt%) and

molybdenum (Mo, 3.5%, wt%) compared to the cells interior (17.1% for Cr, and 2.1% for Mo). To dissolve this micro-segregation and promote complete recrystallization of this sample, we apply laser remelting. As schematically shown in Fig. 1a, the heat generated by each liquid melt pool flows into the surrounding solid material causing an intrinsic HT. Remelting effectively doubles the occurrence of heat affected zones throughout the build, promoting solid-state diffusion of the micro-segregated solute elements. The same sample produced using $h$ = 10 μm but after laser remelting exhibits a more homogeneous elemental distribution across the microstructure (Fig. 3d). We measure

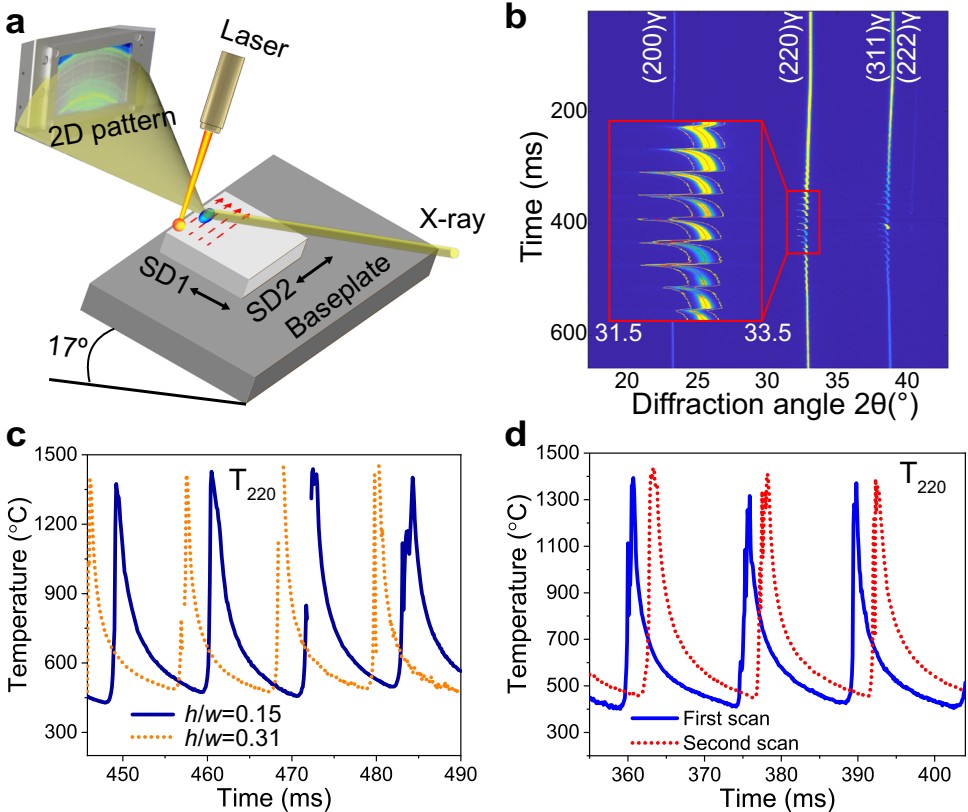

**Fig. 4 | *Operando* X-ray diffraction during laser powder bed fusion of stainless steel 316L (SS316L). a** Schematics of *operando* experiment in reflection mode. SD represents the laser scan direction. **b** Evolution of the diffraction pattern over time during laser scanning of a single layer of SS316L. **c** Temperature profile as a function of time for two different ratios between hatch spacing ($h$) and melt pool width ($w$) corresponding to $h = 20 \, \mu m$ and $h = 40 \, \mu m$ (for $w \approx 130 \, \mu m$). **d** Temperature profile associated with the first and second laser scanning of a layer. Measurements in (**c**) and (**d**) are computed based on the (220) diffraction peak shift. The temperature profiles in (**c**) and (**d**) have been shifted in time for visualization purposes.

the corresponding reduction in the concentration of Cr and Mo at cell boundaries (relative to that within cells) to be 67% (from 2.7% to 0.9%) and 50% (from 1.4% to 0.7%), respectively. As a result, this sample undergoes complete recrystallization after 30 mins at 1050 °C (Fig. 1c).

To investigate how the H and L strategies change the thermal history of the alloy, we carried out *operando* X-ray diffraction experiments using a miniaturized LPBF system installed at the Swiss Light Source (see "Methods" and Fig. 4a). The material's expansion/contraction cycles induced by each melt pool result in periodic changes of the atomic lattice spacing, which can be quantified by tracking the position of the X-ray diffraction peaks generated by the crystal (Fig. 4b). This information allows determining the evolution of lattice strains and, by knowing the thermal expansion coefficient of the material, compute the temperature of the intrinsic HT.

Figure 4c, d plot differences in temperature profile when varying $h/w$ and when employing laser remelting. The comparison reveals no significant effects on the heating and cooling of the alloy. Based on this evidence, we rule out the possibility that the changes in GNDs associated with a different $h$ value stem from variations in cooling rate. The higher GND content is to be ascribed to the larger cumulative plastic strain induced in the material when employing a small $h$ value.

We also conclude that the dissolution of micro-segregation when employing laser remelting results from enhanced solid-state diffusion of solute elements, and not from differences in the alloy solidification path. Indeed, Fig. 4b shows only diffraction peaks coming from the austenitic phase of SS316L throughout multiple melting events across the same layer. Moreover, STEM-EDS analysis of lamellae taken from a thin-wall sample produced during transmission *operando* experiments (Supplementary Fig. 5a) shows similar trends to those we report in

Fig. 3c, d. This *operando* experiment allows quantifying the intrinsic HT across multiple layers during LPBF (Supplementary Fig. 5b). By inputting the temperature profiles obtained experimentally into the DICRA module of Thermo-Calc, we simulate the extent of solid-state diffusion of solute atoms in this specific sample and find good agreement with the experimental results (see "Methods" and Supplementary Fig. 5c and d). This finding confirms that chemical homogenization is a by-product of the intrinsic HT.

Finally, we note that recrystallization has been linked to other microstructural features that are typically found in alloys produced by LPBF, including residual stresses[22]—which supposedly increase the propensity to undergo recrystallization—and oxide nanoparticle formation[23]—which pins the growth of recrystallized grains. We assess the first by means of neutron diffraction (Supplementary Fig. 6) and the second by STEM (Supplementary Fig. 7). We find that a larger $h$ value generates higher residual stresses but yields a material with higher thermal stability compared to one produced with smaller $h$ value. We also find that SS316L produced by laser remelting contains coarser particles but is more prone to undergo recrystallization. Based on these contrasting results, we conclude that neither feature plays a significant role on the thermal stability of our SS316L samples.

**Tailoring mechanical properties via architected microstructure**
To showcase that programming the thermal stability of alloys during LPBF allows engineering the mechanical performance of the material to a greater extent compared to conventional thermo-mechanical processes, we produced samples that integrate different combinations of recrystallized and non-recrystallized microstructure using our strategies (Fig. 5a). Materials with such a microstructure heterogeneity

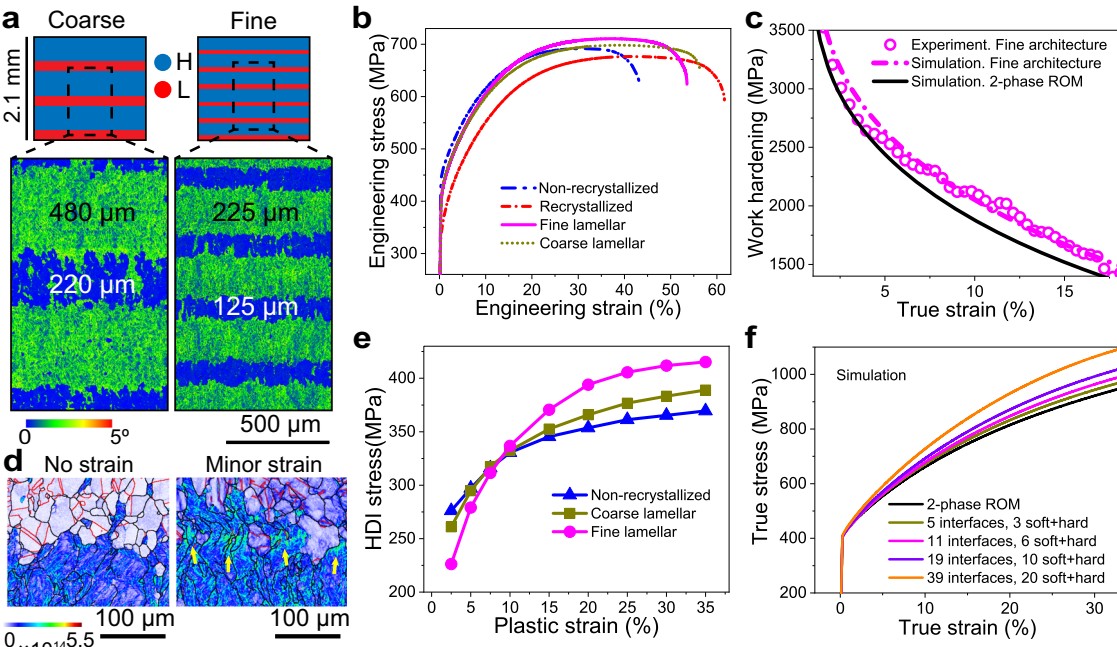

**Fig. 5 | Tensile behavior of layered microstructures. a** Print design and corresponding EBSD KAM maps of coarse and fine layered microstructures.
**b** Experimental stress–strain curves obtained by testing the non-recrystallized, fully recrystallized, and layered microstructures. **c** Comparison between experimental and estimated work hardening rates of the fine architecture. The estimates are based on rule-of-mixture (ROM) using a viscoplastic model. The experimental data deviates from the 2-phase ROM, but matches well that which considers the interface between the constituent microstructures as a third phase with finite thickness and different mechanical properties. **d** Geometrically necessary dislocation (GND) maps measured by EBSD showing pile-ups (the yellow arrows) at the interface between the two microstructure constituents upon mechanical testing. **e** Hetero-deformation induced (HDI) stress evolution of the different samples as a function of plastic strain. **f** Simulated stress–strain curves from layered microstructures with variable number of interfaces using the 3-phase ROM viscoplastic model.

have already been shown to exhibit superior mechanical properties compared to their monolithic counterparts[24,25]. The higher performance stems from the additional strengthening generated by the interfaces between the constituent microstructures—one 'soft' and one 'hard'—and is referred to as hetero-deformation induced (HDI) strengthening[26]. The higher the difference in strength between the two constituents, and the higher the number of interfaces, the larger the resulting HDI contribution.

As reported in Fig. 5b and Supplementary Table 1, our recrystallized SS316L microstructure exhibits a yield strength (YS) of 322 MPa and undergoes uniform elongation (UE) up to 46% before necking. By contrast, the YS and UE of the non-recrystallized microstructure are 440 MPa and 32%, respectively. Figure 5b reports the tensile behavior of the layered microstructures shown in Fig. 5a. In both samples, we keep the volume fraction of the constituent microstructure-layers similar (within a 5% difference) but vary the microstructure-layer thickness (reported in Fig. 5a) to create coarse and fine architectures which contain less or more interfaces, respectively. We find that the YS of both coarse and fine architectures obey rule-of-mixture (ROM), as if the samples were made of a composite material consisting of two phases. This finding implies that the mechanical properties of metals may be tuned to a great level of detail by varying the volume fraction of the constituent microstructures. However, the fine architecture reaches an ultimate tensile strength (UTS) of 708 MPa, which is ~20 MPa higher than that of the upper bound (non-recrystallized) microstructure. This result stems from the higher work hardening exhibited by this architecture, which deviates from the ROM when considering a 2-phase material (Fig. 5c).

Through a combination of experiments and modeling, we prove that this behavior is the product of HDI strengthening and quantify its contribution in our layered microstructures. Upon deformation, we observe extra GND pile-ups along the interface between the non-recrystallized and recrystallized constituents (Fig. 5d). These GNDs accommodate the plastic strain incompatibility between the soft and hard constituents[27] and generate back stresses into the first and forward stresses into the second[28], which contribute to the extra work hardening. We quantify the magnitude of these stresses—which are referred to as HDI stress—through loading-unloading tensile tests of our samples (Supplementary Fig. 8)[26] and plot them in Fig. 5e. The results show that a higher number of interfaces in the architecture yields a more pronounced increase in HDI stress upon mechanical testing. This result stems from an enhanced GND pile-up along the interfaces.

In an effort to develop tools that could be used to design such architectures, we build a viscoplastic model to predict the mechanical response of our SS316L layered microstructures[29,30]. The model (detailed in "Methods" and shown in Supplementary Fig. 9) relies on representative volume elements of hard, soft, and a third pseudo-phase that describes the region surrounding the interface between the two microstructures[27]. As shown in Supplementary Fig. 2e, both recrystallized and non-recrystallized layers exhibit a similar, relatively weak crystallographic texture along the loading direction, which is unlikely to affect HDI strengthening. We fit the experimental stress–strain data of the non-recrystallized and recrystallized microstructures (Fig. 5b and Supplementary Fig. 9b) to define the properties of the first two phases, respectively. By fitting the model to the experimental data, we account for the layer-specific microstructural features, such as the grain size, grain morphology, grain boundary character distribution, texture, and dislocation density. We then estimate the width of the interface zone (IZ) as 15 μm[12] and infer its properties by fitting the experimental stress–strain curve of the coarse architecture (see "Methods" and Supplementary Table 5). Figure 5c and Supplementary Fig. 10a confirm that the model predicts the work hardening of the fine architecture satisfactorily when varying the

number of IZs (11 here) and the volume fraction of the microstructure constituents. This result also suggests that IZ properties and thickness do not vary with microstructure architecture. Thus, we use the visco-plastic model to predict the expected behavior of other hypothetical layered microstructures with higher number of IZs (Fig. 5f).

The theoretical limit to this type of architecture is when the IZ thickness equals that of the individual microstructure-layers. The more practical limit, however, is set by the spatial resolution of LPBF machines (of the order of 100 μm along the build direction in this work), which is a function of melt pool size and laser parameters employed. When attempting to produce an architecture with 19 interfaces, we find that the recrystallized microstructure expands beyond the initial design, yielding connected regions of recrystallized material that limit the maximum attainable strength (Supplementary Fig. 10b).

## Discussion

In this work, we focus on the capability to engineer the mechanical properties of SS316L using our LPBF strategies to program the thermal stability of SS316L and create layered microstructure architectures. We use these results as a demonstration of the potential of our microstructure control. However, the differences in crystallographic texture, grain structure, and grain boundary character distribution which result from recrystallization may also inspire other microstructure designs that lead to superior performance or novel functionalities. Programmable, site-specific recrystallization could be used to optimize materials resistance against failure as a result of fatigue[31] or hydrogen embrittlement[32], for instance. In that regard, we expect our strategy to be applicable to many other materials since it relies on the control of microstructural features that are ubiquitous in metal alloys produced by LPBF—namely dislocations accumulation and elemental micro-segregation upon solidification. Our work opens the path to designing metal parts that exhibit microstructure-optimized properties along-side the widely celebrated topology-optimized performance brought by shape complexity.

## Methods

### Sample production

We printed all SS316L samples with the exception of that shown in Fig. 1d using a custom-made LPBF machine, equipped with a 100 W continuous wave IPG fiber laser source of 1070 nm wavelength. We used gas atomized SS316L powder with nominal composition as shown in Supplementary Table 2. The particle size ranged from 5 μm to 25 μm. The laser beam size at focus was of 25 μm and all prints were carried out in nitrogen atmosphere. We printed a first batch of cubic samples (8 × 8 × 8 mm³) for microstructure characterization, to investigate the effects of process parameters on the alloy's thermal stability. For mechanical testing, we printed 24 mm long, 5 mm wide, and 10 mm tall specimens, which we cut into dog-bone-shaped plates. All samples were produced using the same laser power of 60 W, scanning speed of 600 mm/s, scan rotation of 90°, and powder layer thickness of 10 μm. We instead varied the hatch spacing $h$ from 10 μm to 35 μm as well as the number of times the laser was scanned across each layer (as shown in Supplementary Table 3). By such laser parameters, the averaged melt pool width, $w$, is ~70 μm. By "remelting" we imply that the laser scanned twice on the specific region within the layer, effectively melting that area twice. When producing multiple samples within the same LPBF build, we carried out remelting on one sample at the time, just like in a normal single-scan LPBF process. In these cases, the time it took to complete one entire layer (i.e., after scanning all samples) varied from 1 to 3 min, which was long enough for the scanned areas to cool down without introducing significant thermal buildups. All samples produced using these parameters were 99.9% dense, as measured through optical microscopy. We selected the H and L strategies from within this processing window.

To produce the sample with the binary code standing for "AM" (Fig. 1d), we used a different LPBF machine and processing parameters, but similar H and L strategies. The machine was equipped with a 200 W SPI fiber laser source with wavelength of 1060 nm. The laser spot size on the build plate was adjusted to be 100 μm in diameter. We used constant laser power of 200 W, scanning speed of 650 mm/s, scan rotation of 90°, and layer thickness of 25 μm. The averaged melt pool width measured from optical micrographs is ~230 μm. We set $h$ = 100 μm to bestow the microstructure with high thermal stability. We instead used $h$ = 50 μm and applied laser remelting to program recrystallization to occur upon HT.

To investigate the thermal stability of all samples, we employed HTs of 30 min at variable temperatures between 1050 °C and 1200 °C using an Elite laboratory chamber furnace. Samples were then cooled down in air.

### Electron microscopy characterization

We prepared the cube samples for microstructure analysis following standard metallographic procedures. We assessed the microstructure using a JOEL JSM-7800F Prime field emission scanning electron microscope (SEM) equipped with an electron backscatter diffraction (EBSD) detector (Oxford Instrument, Symmetry). We acquired EBSD measurements using a step size 0.5 μm for as-built, heat-treated, and tensile samples. We employed a step size of 5 μm step size for the sample shown in Fig. 1d, and of 1.5 μm for the samples shown in Figs. 1e and 5a and Supplementary Fig. 10. We analyzed the EBSD data using the software AZtecCrystal (by Oxford Instruments) and MTEX 5.6, which is a comprehensive MATLAB toolbox for analyzing and plotting crystallographic quantities[33]. We classified GBs according to their misorientation into low-angle grain boundaries (LAGBs, from 2° to 15°), high-angle grain boundaries (HAGBs, >15°), and twin boundaries (TBs, 60° about <111>). To reveal the solidification structure of SS316L produced by LPBF, we etched the polished samples in a bath of hydrofluoric and nitric acid (HF:HNO₃:H₂O = 1:4:45) for 20 min.

To estimate relative changes in GND density from EBSD measurements, we used strain gradient theory[27,34]:

$$\rho = \frac{2\theta}{Xb}. \tag{2}$$

Here, $\theta$ is the average local misorientation angle measured in KAM maps, $X$ corresponds to the scan step-size (0.5 μm for this analysis), and $b$ is the magnitude of the Burgers vector.

To characterize the solidification structure, we relied on bright-field (BF) and high-angle annular dark-field (HAADF) imaging using a JEM-GrandARM aberration-corrected transmission electron microscope (TEM) operated at 300 kV in scanning transmission electron microscopy (STEM) mode. We carried out STEM energy-dispersive X-ray spectroscopy (EDS) to map the elemental micro-segregation at cell boundaries. TEM lamellae were cut from the etched sample surface by focused ion beam (FIB) using a gallium ion source on a ZEISS Crossbeam 540. To enable a direct comparison between solidification structures taken from different samples, we lifted the TEM lamellae from grains with similar crystallographic orientation (<110> parallel to the build direction and <001> along the nominal vector of the surface) and at a similar distance from the melt pool fusion boundaries. The lamellae used in Fig. 3c, d were cut from the top layer of SS316L cubes that underwent no remelting and one remelting, respectively. The ones used in Supplementary Fig. 5c, d were deliberately cut from the top layer and from 90 μm below the surface of a SS316L thin wall sample.

### Neutron diffraction

We assessed the residual stresses in our cube samples at the Australian Nuclear Science and Technology Organization (ANSTO), using the stress diffractometer KOWARI. We selected a nominal wavelength of

1.53 Å and relied on (311) reflections, which are found at a 2θ angle of ~90° in FCC austenitic stainless steel. We collected the diffraction patterns and assessed the strain profiles from a cube-shaped 1 × 1 × 1 mm³ neutron gauge volume. The position of each sample in the diffractometer was determined from the intensity change of the diffracted signal, which ensures a positioning accuracy better than 0.1 mm.

The measurements consisted of a 7 × 7 grid of points equally spaced by 1 mm along the building direction (BD) and scanning direction (SD). We fitted the neutron diffraction profiles using a Gaussian model to determine the peak position. The resulting target strain accuracy was 50 μstrain, which translates into a stress accuracy of ~15 MPa. The elastic strains ($\varepsilon$) in the three orthogonal directions vertical (VD, parallel to BD), transversal (TD, parallel to SD1), and longitudinal (LD, parallel to SD2) were converted into the residual stresses ($\sigma$), $\sigma_{VD}$, $\sigma_{TD}$, and $\sigma_{LD}$ using the generalized Hooke's law (Supplementary Fig. 6). We used an elastic constant, $E$, and Poisson's ratio, $\nu$, of 183.5 GPa and 0.31, respectively, for the (311) plane[35].

### *Operando* X-ray diffraction

We employed a miniaturized LPBF setup, which is installed on the microXAS beamline located at the Swiss Light Source. Details on the setup can be found in a previous study[36]. The *operando* measurements were carried out using a 17.2 keV X-ray beam, which was focused down to 34 × 20 μm² (full-width at half-maximum) with Kirkpatrick–Baez mirrors. During laser scanning, we captured X-ray diffraction patterns through an in-house developed EIGER1M detector[37] at a frequency of 20 kHz. We synchronized the X-ray detector and laser operation using a hardware trigger.

We performed two types of *operando* experiments, in reflection and in transmission, which we schematically show in Fig. 4a and Supplementary Fig. 5a, respectively. To investigate the effect of hatch spacing on the thermal history of the alloy, we printed two 4 × 4 mm² rectangular builds using a laser energy of 175 W, a laser spot size of 43 μm, a scanning speed of 400 mm/s, a powder layer thickness of 30 μm, and variable hatch spacings of 20 μm and 40 μm. Under such process parameters, the measured melt pool width, $w$, is ~130 μm. To investigate the effect of laser remelting, we produced a third 4 × 4 mm² rectangular build using a power of 100 W, a scanning speed of 200 mm/s, a powder layer thickness of 30 μm, and a hatch spacing of 40 μm. We compared the thermal profiles from two hatch spacings (shown in Fig. 4c), powder melting (first laser scan), and the remelted surface (shown in Fig. 4d) by analyzing the evolution of the X-ray diffraction patterns acquired in reflection mode. In this mode, the X-ray beam was focused onto the surface under an incidence angle of 17°, resulting in a projected illuminated area of 34 × 68 μm² (see Fig. 4a).

The second series of experiments aimed at investigating the thermal cycles the microstructure beneath the melting area was subjected to. To this end we first produced an 8 mm long, 0.1 mm wide, and 4 mm tall wall using a laser power of 100 W, a spot size of 43 μm, and a scanning speed of 200 mm/s. After removing the surrounding powder, we focused the X-ray beam on the wall cross-section under an incidence angle of 6°, at various depths below the top surface (see Supplementary Fig. 5a). We then acquired X-ray diffraction patterns in transmission mode as we scanned the laser on the top surface.

The *operando* measurements were calibrated using a NaCl standard, knowing the exact sample-to-detector distance, beam center, and detector tilt. We integrated the 2D patterns using the software package pyFAI. Single peak profile analysis was performed using an in-house written Matlab routine. We determined lattice strain from the relative change in the diffraction peak position. We then converted this values into temperature using the thermal expansion coefficients found in Touloukian's book[38]. Finally, we computed the resulting cooling rates using the derivative of the temperature-time curves[39]. The time resolution for temperature profiles shown in Fig. 4 and Supplementary Fig. 5 is 50 μs.

We employed the temperature profiles obtained from the *operando* experiments in transmission to simulate the elemental diffusion of Cr and Mo driven by the intrinsic HT during laser remelting (see Supplementary Fig. 5). The simulations were conducted using the DICTRA (Diffusion Controlled TRAnsformation) module[40] of the Thermo-Calc (2020a) software package, and employing the thermodynamic (TCFE9) and mobility (MOBFE5) databases developed for steels and Fe-alloys.

### FEM simulations

To investigate the cumulative plastic strain induced in the alloy during the LPBF process as a function of laser parameters, we set up FEM simulations using the ABAQUS additive manufacturing module. All materials parameters used in the simulation are reported in Supplementary Table 4. We simulated an eight-layer LPBF print process, where we adopt progressive element activation to model the powder deposition process and a unidirectional laser scanning strategy rotated by 90° each layer (Supplementary Fig. 3a). Once completed, the simulation included a cooling down phase to room temperature. We first carried out a transit heat transfer analysis to obtain the temperature profile during the printing and cooling steps, which we subsequently passed to a static structural analysis to obtain the cumulative plastic strain in the sample. We modeled the laser heat source using a Goldak distribution[41]. The resulting melt pool width, $w$, in the current simulation was around three elements wide (Supplementary Fig. 3b). The effective plastic strain accumulation induced by thermal expansion/contraction cycles is modeled with a perfectly plastic model upon solidification. We performed a series of FEM simulations with varying hatch spacing, $h$, from 2/3 $w$ to 1/12 $w$. We treated remelting stemming from the overlapping of adjacent melt pools by resetting the effective plastic strain in the overlapping region to zero.

### Viscoplastic model

We developed an J2 viscoplastic model using a representative volume element method[27,42] to account for the extra work-hardening stemming from HDI. All model parameters used in this work are reported in Supplementary Table 5. The effective plastic strain rate, $\dot{\epsilon}_p$, is related to the effective stress $\sigma_e$ by:

$$\dot{\epsilon}_p = \Phi \left( \frac{\sigma_e}{\sigma(\epsilon_p)} \right)^{\frac{1}{m}}, \tag{3}$$

Here, $\Phi$ is the characteristic strain rate and $m$ is a constant that dictates the strain rate sensitivity of the constituent phases. We set $\Phi$ to $10^{-3}\,s^{-1}$ for all phases, which corresponds to the strain rate we employed during all tensile tests. We chose 0.005 for $m$ to ensure weak rate-dependence that is representative of quasi-static loading conditions. The flow resistance, $\sigma(\epsilon_p)$, of the individual phases is described by a five-parameter work hardening function:

$$\sigma(\epsilon_p) = \sigma_0 + Q\left(1 - \exp(-k_1 \epsilon_p)\right) + H\epsilon_p^{k_2}, \tag{4}$$

where $\sigma_0$ is the initial flow stress (i.e., the measured yield stress), the second term accounts for the steep work hardening at the early yielding stage, and the third term accounts for the saturated hardening at larger strain. $Q$, $k_1$, $H$, $k_2$ are model parameters that define the strain hardening behavior of SS316L produced by LPBF. $Q$ is the transient flow stress, $k_1$ is the transient hardening constant, $H$ is the strain hardening rate, and $k_2$ is the hardening exponent. These material parameters for both non-recrystallized and recrystallized microstructures were determined by minimizing the L2 error between the simulated stress–strain curves and the corresponding experimental stress–strain data using the Nelder-Mead simplex search algorithm[43]. Supplementary Fig. 9b confirms a good agreement between the two.

During deformation of the layered microstructures, GNDs accumulate at the interfaces between the two constituent microstructures to accommodate the plastic strain gradient. This leads to the formation of the IZ. The width of the IZ in this work is estimated by[12]:

$$l_{IZ} \approx \left(\frac{\mu}{\sigma_y}\right)^2 b, \tag{5}$$

where $\mu$ is the shear modulus[44], $\sigma_y$ is the yield stress of the recrystallized microstructure, and $b$ is the length of the burgers vector[45]. The resulting width of 15 μm is comparable to the average grain size of the recrystallized phase[30]. The plastic behavior of the IZ is described by:

$$\dot{\epsilon}_p = \Phi \left(\frac{|\sigma_e - \sigma_b|}{\sigma_{IZ}(\epsilon_p, \eta)}\right)^{\frac{1}{m}}, \tag{6}$$

where $\sigma_b$ is the effective back stress, $\sigma_{IZ}$ is the flow resistance as a function of the accumulated plastic strain $\epsilon_p$, and $\eta = \sqrt{\epsilon_{p,i}\epsilon_{p,i}}$ is the effective strain gradient across the interface. A mechanism-based hardening function[42] is adopted to evaluate $\sigma_{IZ}$:

$$\sigma_{IZ}(\epsilon_p, \eta) = \sigma_0 \sqrt{f(\epsilon_p) + \alpha\eta}, \tag{7}$$

where $f(\epsilon_p)$ represents the conventional isotropic hardening (see Eq. (4)) and $\alpha\eta$ represents the extra hardening due to the plastic strain gradient. $\alpha$ represents the intrinsic material length scaling with strain gradients in strain gradient plasticity theories[42]. The back stress rate is expressed as:

$$\dot{\sigma}_b = c(\beta\eta - \sigma_b)\dot{\epsilon}_p. \tag{8}$$

Here $\beta\eta$ represents the saturated back stress, which increases with the strain gradient between and soft and hard phases. $\beta$ is a material parameter that determines the build-up of back stresses due to the plastic strain gradient, and $c$ is the model coefficient controlling the back stress increasing rate. Here, we assumed a linear relationship between the saturated back stress and $\eta$, as well as between $\dot{\sigma}_b$ and $\dot{\epsilon}_p$, for simplicity. This is consistent with previous works[29]. In the current model, deformation inside the IZs is assumed to be homogenous, which is a valid assumption when the width of the IZs is much smaller than the width of the recrystallized lamella.

As shown in Supplementary Fig. 9c, we adopt a one-dimension model where only the stress component $\sigma_{11}$ is non-zero and varies in different phases along direction perpendicular to layer orientation. We compute the stress of the layered microstructure as the sum of the stresses in each phase, weighted by the phase volume fraction. We set the material parameters for isotropic hardening in IZs ($\sigma_0$, $Q$, $k_1$, $H$, $k_2$) to be the same as those of the soft phase, since IZs are assumed to form in the recrystallized microstructure. We determine the model parameters $\alpha$, $\beta$, and $c$ associated with the strain gradient plasticity in IZs by minimizing the L2 error between the simulated and experimental stress–strain curve of the coarse architecture. We then use the same parameters when simulating the other architectures.

### Tensile testing
We cut dog-bone-shaped plates with a gauge length of 7 mm and a width of 2.1 mm (Supplementary Fig. 9a) from the tensile specimens produced by LPBF using electric discharge machining. The selected width ensures that the gauge area contains 11 and 5 IZs between different microstructure-layers in the fine and coarse architecture, respectively. We carry out uniaxial tensile tests on a SHIMADZU AG-X plus machine equipped with a load cell of 10 kN and a TRViewX video extensometer. We set the tensile strain rate to 0.001 s⁻¹ and the load

direction parallel to the lamellae orientation. We repeat each tensile test at least three times for each type of microstructure.

To compare the HDI stress evolution in different architectures, we conducted loading-unloading cycles at various plastic strains during tensile test on the same mechanical machine. Upon loading to a specific strain level, the samples were unloaded in a force-control mode to 20 N at a rate of 20 N/s before being re-loaded to the following strain level (Supplementary Fig. 8). We used strain levels of (2.5%, 5%, 7.5%, 10%, 15%, 20%, 25%, 30%, and 35%).

## Data availability
All data supporting the findings of this study are available from the corresponding author upon request. Source data are provided with this paper.

## Code availability
The code used for modeling and simulation in this work are available from the corresponding author upon request.

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

## Acknowledgements

The authors from Nanyang Technological University (NTU) and University of Cambridge would like to acknowledge M.J. Demkowicz, Z. Cordero, and M. Duchamp for valuable discussion, C. Todaro for the beamtime of neutron diffraction in ANSTO, Z. Wang and T.P. Le for technical support, and the Facilities for Analysis, Characterization, Testing and Simulations (FACTS) at NTU for access to electron microscopy equipment. This research was funded by the National Research Foundation (NRF) Singapore, under the NRF Fellowship program (NRF-NRFF2018-05). S.V.P. acknowledges support from the Swiss National Science Foundation (SNF Sinergia 193799). H.L.S. acknowledges support from the Science and Engineering Research Council, Agency for Science, Technology and Research (A*STAR), Singapore (142 68 00088). H.G. acknowledges support from Advanced Models for Additive Manufacturing (AM²) program under A*STAR (M22L2b0111) and support as a Distinguished University Professor of NTU and Scientific Director of Institute of High Performance Computing of the A*STAR, Singapore.

## Author contributions

S.Ga and M.S. conceived the work, designed the experiments, and drafted the initial manuscript. S.Ga produced all samples, performed microstructure characterization, carried out mechanical tests, and conducted data analysis. Z.L. and H.G. developed the viscoplastic model and conducted theroetical analysis. S.V.P., S.Go, D.F.S., and H.V.S. carried out the operando X-ray experiments and interpreted the data. J.G. and J.V.V. performed the STEM charaterization and data analysis. V.L. conducted the measurement of residual stresses by neutron diffraction and analyzed the data. Z.H. and H.L.S. guided the LPBF process and conducted the simulation of elemental difusion using Thermal-Calc. H.V.S., H.G., and M.S. supervised the work. All co-authors contributed to data discussion and the revision of the initial manuscript.

## Competing interests

The authors declare no competing interests.
