## [Peer Review File · Nature Communications]

Additive manufacturing of alloys with programmable microstructure and propertiesReviewers' Comments:

Reviewer #1:

Remarks to the Author:

Gao et al. provide a very interesting and comprehensive study on local tailoring of the microstructure in 316L stainless steel. The authors apply varying processing parameters (hatch distance and laser re-scanning) to manipulate the local dislocation density that is used as a driving force to promote recrystallization upon subsequent heat treatment. The correlation between processing conditions, microstructure evolution, including the local thermal stability, and mechanical properties is convincingly presented. The authors also demonstrate their 'microstructure programming' by writing a binary code in the material's structure and by producing layered specimens with adjustable mechanical properties.

The research study is certainly sound. The results and methods are presented in a clear and reproducible way. It would be helpful to also provide the content of trace elements in the chemical analysis presented in Supplementary Table 1. Furthermore, it is not clear how the local crystallographic texture within the individual layers is treated in the plasticity model and if the texture influences the mechanical properties. The conclusions are well supported by the research results and the work does contain novel aspect of metal AM.

Referring to the latter, the manuscript is mostly a combination of individual aspects that have in one or another way been published before. The possibility to manipulate local microstructures within the processing limits of AM (here LPBF) has been reported by multiple research groups, e.g. writing 'DOE' in EBM-processed IN718 [1]. Also, the fact that the recrystallization behaviour during post-treatment can be influenced by varying the solidification behaviour during AM has been researched before and appears quite intuitive from a metal physics point of view (as also investigated by the authors for the same material [2]). The novelty of modifying the dislocation density and thermal stability by varying the hatch distance and applying laser re-scanning is appreciated. Unfortunately, the presented approach appears rather as a trial-and-error approach including a phenomenological discussion. There is no attempt to further describe the evolution of the dislocation density in a physics-based manner and neither a related modelling approach, which would be a major contribution to the field of research. Keeping this in mind, the overall study, which is certainly of high quality, is rather a feasibility study that demonstrates what is already know in metal AM. The combination of 'hard' and 'soft' layers and the related influence on the deformation behaviour (hetero-deformation induced strengthening after [3]) is also known from other heterogeneous materials. The plasticity modelling approach does hardly contribute to a better understanding of the material behaviour and rather distracts from the actual focus of the work.

[1] R. R. Dehoff, M. M. Kirka, W. J. Sames, H. Bilheux, A. S. Tremsin, L. E. Lowe and S. S. Babu: 'Site specific control of crystallographic grain orientation through electron beam additive manufacturing', *Mater. Sci. Technol.*, 2015, 31, (8), 931–938.

[2] Gao, S. et al. Recrystallization-based grain boundary engineering of 316L stainless steel produced via selective laser melting. *Acta Materialia* 200, 366-377 (2020).

[3] Zhu, Y. & Wu, X. Heterostructured materials. *Progress in Materials Science* 131 (2023). <https://doi.org/10.1016/j.pmatsci.2022.101019>.

Reviewer #2:

Remarks to the Author:

The authors present the results of a detailed experimental program, backed up with some modelling to help in the interpretation of the results, that seeks to manipulate the solidification behaviour of a well understood stainless steel to allow the "programming" of the microstructure to exhibit differing degrees of recrystallisation upon heat treatment and hence precise control of local mechanical properties on a length scale hitherto impossible in additive manufacture. The work which includes extensive in-operando studies of the process is extremely well conducted and the finding are very

interesting indeed – I would consider that the paper would be of great interest to the AM community particularly but beyond that to the metallurgical community as a whole.

There is one aspect of the methodology that I found confusing however, and this relates to the “re-scanning” employed in the L-Strategy. I could not quite work out whether re-scanning was in fact remelting. It would seem that it was but this isn’t made explicitly clear in the paper. If re-scan means remelting – then this should be made clear also is each track of the hatch remelted n times before moving to the next or is the whole L-Strategy region melted / scanned then “rescanned” n times? Also – what is the time delay between each scan? The extent of the retained heat in the material locally will depend, critically, on the scanning strategy used. I would like this discussed more fully either in the methods or in the supplementary materials, because as it stands the method lacks sufficient detail to be replicated by a third party.

Other than this minor quibble, the paper is very interesting and would find a ready audience. Its exploration of GNDs and recrystallisation to manipulate microstructure is a welcome contribution.

Reviewers' comments

Authors' responses Reviewers' Comments:

Reviewer #1:

Remarks to the Author:

Gao et al. provide a very interesting and comprehensive study on local tailoring of the microstructure in 316L stainless steel. The authors apply varying processing parameters (hatch distance and laser re-scanning) to manipulate the local dislocation density that is used as a driving force to promote recrystallization upon subsequent heat treatment. The correlation between processing conditions, microstructure evolution, including the local thermal stability, and mechanical properties is convincingly presented. The authors also demonstrate their 'microstructure programming' by writing a binary code in the material's structure and by producing layered specimens with adjustable mechanical properties.

The research study is certainly sound. The results and methods are presented in a clear and reproducible way. It would be helpful to also provide the content of trace elements in the chemical analysis presented in Supplementary Table 1. Furthermore, it is not clear how the local crystallographic texture within the individual layers is treated in the plasticity model and if the texture influences the mechanical properties. The conclusions are well supported by the research results and the work does contain novel aspect of metal AM.

Referring to the latter, the manuscript is mostly a combination of individual aspects that have in one or another way been published before. The possibility to manipulate local microstructures within the processing limits of AM (here LPBF) has been reported by multiple research groups, e.g. writing 'DOE' in EBM-processed IN718 [1]. Also, the fact that the recrystallization behaviour during post-treatment can be influenced by varying the solidification behaviour during AM has been researched before and appears quite intuitive from a metal physics point of view (as also investigated by the authors for the same material [2]). The novelty of modifying the dislocation density and thermal stability by varying the hatch distance and applying laser re-scanning is appreciated. Unfortunately, the presented approach appears rather as a trial-and-error approach including a phenomenological discussion. There is no attempt to further describe the evolution of the dislocation density in a physics-based manner and neither a related modelling approach, which would be a major contribution to the field of research. Keeping this in mind, the overall study, which is certainly of high quality, is rather a feasibility study that demonstrates what is already known in metal AM. The combination of 'hard' and 'soft' layers and the related influence on the deformation behaviour (hetero-deformation induced strengthening after [3]) is also known from other heterogeneous materials. The plasticity modelling approach does hardly contribute to a better understanding of the material behaviour and rather distracts from the actual focus of the work.

[1] R. R. Dehoff, M. M. Kirka, W. J. Sames, H. Bilheux, A. S. Tremsin, L. E. Lowe and S. S. Babu: 'Site specific control of crystallographic grain orientation through electron beam additive manufacturing', *Mater. Sci. Technol.*, 2015, 31, (8), 931–938.

[2] Gao, S. et al. Recrystallization-based grain boundary engineering of 316L stainless steel produced via selective laser melting. *Acta Materialia* 200, 366-377 (2020).

[3] Zhu, Y. & Wu, X. Heterostructured materials. *Progress in Materials Science* 131 (2023). <https://doi.org/10.1016/j.pmatsci.2022.101019>.

Reviewer #2:

Remarks to the Author:

The authors present the results of a detailed experimental program, backed up with some modelling to help in the interpretation of the results, that seeks to manipulate the solidification behaviour of a well understood stainless steel to allow the "programming" of the microstructure to exhibit differing degrees of recrystallisation upon heat treatment and hence precise control of local mechanical properties on a length scale hitherto impossible in additive manufacture. The work which includes extensive in-operando studies of the process is extremely well conducted and the findings are very interesting indeed – I would consider that the paper would be of great interest to

the AM community particularly but beyond that to the metallurgical community as a whole.

There is one aspect of the methodology that I found confusing however, and this relates to the “re-scanning” employed in the L-Strategy. I could not quite work out whether re-scanning was in fact remelting. It would seem that it was but this isn’t made explicitly clear in the paper. If re-scan means remelting – then this should be made clear also is each track of the hatch remelted n times before moving to the next or is the whole L-Strategy region melted / scanned then “rescanned” n times? Also – what is the time delay between each scan? The extent of the retained heat in the material locally will depend, critically, on the scanning strategy used. I would like this discussed more fully either in the methods or in the supplementary materials, because as is stands the method lacks sufficient detail to be replicated by a third party.

Other than this minor quibble, the paper is very interesting and would find a ready audience. Its exploration of GNDs and recrystallisation to manipulate microstructure is a welcome contribution.

Reviewer #1

Gao et al. provide a very interesting and comprehensive study on local tailoring of the microstructure in 316L stainless steel. The authors apply varying processing parameters (hatch distance and laser re-scanning) to manipulate the local dislocation density that is used as a driving force to promote recrystallization upon subsequent heat treatment. The correlation between processing conditions, microstructure evolution, including the local thermal stability, and mechanical properties is convincingly presented. The authors also demonstrate their ‘microstructure programming’ by writing a binary code in the material’s structure and by producing layered specimens with adjustable mechanical properties. The research study is certainly sound. The results and methods are presented in a clear and reproducible way.

Response: We thank the Reviewer for the very positive feedback on our work, and for acknowledging the soundness of our study. Hereafter, the Reviewer raises some concerns on the novelty of our work and makes an important comment on the depth of our analysis, which we took in great consideration when revising the manuscript. We hope that our responses and revisions to the manuscript will satisfy the Reviewer.

It would be helpful to also provide the content of trace elements in the chemical analysis presented in Supplementary Table 1.

Response: We thank the Reviewer for the suggestion. The chemical composition shown in the initial Supplementary Table 1 was provided by the metal powder supplier. In addition to that, we now include inductively coupled plasma (ICP) measurements which we carried out on the powder used in the work. We report the Table hereafter for convenience:

Cr	Ni	Mo	Mn	Si	Cu
17.26±0.10	12.78±0.04	2.40±0.04	0.79±0.09	0.62±0.01	0.2±0.01
Al	C	P	S	Fe	
0.06±0.01	0.027	<0.01	<0.01	Bal.	

Furthermore, it is not clear how the local crystallographic texture within the individual layers is treated in the plasticity model and if the texture influences the mechanical properties.

Response: The Reviewer makes a good point. We have performed additional EBSD measurements to quantify the crystallographic textures in all our samples, which we now present in the Supplementary Fig. 2 (reported below for convenience). As the grain orientation map and inverse pole figures show in Fig. 2e, both recrystallized and non-recrystallized regions in our layered microstructure sample exhibit a similar—albeit weak—crystallographic texture along the loading direction (i.e., the direction that is parallel to the X-axis in the EBSD map). As such, we do not expect notable differences in the plastic behavior of the two microstructures, especially with regard to HDI strengthening.

Most importantly, we would like to highlight that our viscoplastic model is based on fitting the experimental stress-strain data acquired from tensile tests on the monolithic microstructures (i.e., samples which were either completely recrystallized or non-recrystallized). As such, the model automatically captures the contribution of any inherent microstructural feature in these samples, such as their grain size, grain morphology, grain boundary character, dislocation density, and grain orientation distribution. Therefore, we need not include any specific information on the crystallographic texture of the microstructure constituents in the model, as they are automatically accounted for. On page 16 of the revised manuscript, we clarify this point to avoid any further confusion.

Supplementary Fig. 2. Analysis of the different microstructural features produced through site-specific recrystallization. (a) Grain boundary character distribution in the binary code sample as measured by EBSD. **(b)** Corresponding

inverse pole figure map showing crystal orientation within the build plane. **(c)** Grain boundary character distribution in the layered microstructure sample as measured by EBSD. **(d)** And **(e)** Corresponding inverse pole figure maps showing crystal orientation along the build direction and within the build plane, respectively. In (e) we show the inverse pole figures from the recrystallized and non-recrystallized layers along tensile direction.

The conclusions are well supported by the research results and the work does contain novel aspect of metal AM. Referring to the latter, the manuscript is mostly a combination of individual aspects that have in one or another way been published before. The possibility to manipulate local microstructures within the processing limits of AM (here LPBF) has been reported by multiple research groups, e.g. writing 'DOE' in EBM-processed IN718 [1]. Also, the fact that the recrystallization behaviour during post-treatment can be influenced by varying the solidification behaviour during AM has been researched before and appears quite intuitive from a metal physics point of view (as also investigated by the authors for the same material [2].

Response: We agree with the Reviewer that several other researchers have demonstrated the possibility of using AM (specifically LPBF) to manipulate the microstructure of metal alloys site-specifically. This idea is not new. However, we argue that both our approach to site-specific microstructure control as well as the level of control we achieve are completely different and much more advanced compared to those developed before. Our paper is the first to demonstrate site-specific control over the thermal stability of a metal alloy and, consequently, over several microstructural features at once, including grain morphology, texture, grain boundary character distribution, and dislocation density.

We reinforce this idea in the third paragraph of page 3 and in the second paragraph of page 7 of the revised manuscript. The latter refers the reader to a comprehensive microstructure analysis that supports our claims, which we have added as Supplementary Fig. 2.

We would also like to point out that the ability to trigger recrystallization without having to mechanically deform the microstructure is major novelty of our work. The Reviewer is right to say that previous publications report on the role of the solidification microstructure on recrystallization. However, in all those studies—including ours—recrystallization required an additional level of strain, which could only be provided via mechanical deformation. As we explain in our introduction, this limitation significantly restricts the opportunities to engineer the microstructure and properties of metal parts produced by AM. Our work provides a novel pathway to unlock this level of microstructure engineering without requiring any mechanical deformation, and thus in complete respect of the near-net-shape nature of the LPBF process. We believe that this *new* capability is a significant breakthrough in metal AM.

We would argue that it is certainly a breakthrough in the field of heterostructured materials, where the limited versatility of the thermo-mechanical processes currently employed restricts the design freedom of these materials. We discuss this point in the second paragraph on page 7.

Finally, we also believe that the idea of ‘programmable’ recrystallization is another important and novel aspect of our work, as it may open new opportunities to designing materials with tunable properties. We further elaborate on this point in the Discussion section.

We hope that the explanations we provide in this response will convince the Reviewer that our work is not a mere combination of prior achievements and instead contains multiple novel aspects, beyond those which the Reviewer mentions in the next comment.

The novelty of modifying the dislocation density and thermal stability by varying the hatch distance and applying laser re-scanning is appreciated. Unfortunately, the presented approach appears rather as a trial-and-error approach including a phenomenological discussion. There is no attempt to further describe the evolution of the dislocation density in a physics-based manner and neither a related modelling approach, which would be a major contribution to the field of research.

Response: We thank the Reviewer for this constructive feedback. We agree that our work would benefit from a model that could capture the different dislocation densities induced by our laser scanning strategies. To this end, we have carried out extensive EBSD investigations as well as finite element model simulations to systematically study the dislocation density evolution. We then interpreted these results on the basis of a simple analytical model, which was proposed by Mike Ashby in his seminal paper (reference [16] in the manuscript, “The deformation of plastically non-homogeneous materials”. *The Philosophical Magazine: A Journal of Theoretical Experimental and Applied Physics* 21(170), 399-424 (1970)). These new results confirm our initial theory and findings. We refer the Reviewer to pages 8 and 9 of the revised manuscript to assess this new contribution.

Keeping this in mind, the overall study, which is certainly of high quality, is rather a feasibility study that demonstrates what is already known in metal AM. The combination of ‘hard’ and ‘soft’ layers and the related influence on the deformation behaviour (hetero-deformation induced strengthening after [3]) is also known from other heterogeneous materials. The plasticity modelling approach does hardly contribute to a better understanding of the material behaviour and rather distracts from the actual focus of the work.

Response: We hope that our responses to the Reviewer’s initial comments around the novel aspects of our work, as well as the revisions we made to improve the description and interpretation of the dislocation density evolution will convince the Reviewer of the overall novelty and impact of our work.

We completely agree that the application of our LPBF processing strategies to produce hetero-structured materials is—and was always intended as—a demonstration of the potential of our microstructure control abilities (i.e., a “feasibility study”). One that, we think, provides a solid argument for how our processing

strategies may be used in future engineering applications, and which may inspire other researchers to design novel 'microstructure architectures' by LPBF. On that note, we believe that the viscoplastic model contributes greatly to this end. It provides a benchmark for our experiments, helping identify limitations in spatial resolution and providing quantitative estimates of what mechanical performance a layered material could display. Thus, we decided to keep the model in the revised manuscript.

We have clarified the "feasibility study" nature of this part of the work in the Discussion section of the revised manuscript.

Reviewer #2

The authors present the results of a detailed experimental program, backed up with some modelling to help in the interpretation of the results, that seeks to manipulate the solidification behaviour of a well understood stainless steel to allow the “programming” of the microstructure to exhibit differing degrees of recrystallisation upon heat treatment and hence precise control of local mechanical properties on a length scale hitherto impossible in additive manufacture. The work which includes extensive in-operando studies of the process is extremely well conducted and the findings are very interesting indeed – I would consider that the paper would be of great interest to the AM community particularly but beyond that to the metallurgical community as a whole.

There is one aspect of the methodology that I found confusing however, and this relates to the “re-scanning” employed in the L-Strategy. I could not quite work out whether re-scanning was in fact remelting. It would seem that it was but this isn’t made explicitly clear in the paper. If re-scan means remelting – then this should be made clear also is each track of the hatch remelted n times before moving to the next or is the whole L-Strategy region melted / scanned then “rescanned” n times?

Response: We thank the Reviewer for the very strong endorsement on our work. We also thank the Reviewer for pointing out two possible sources of confusion. Re-scanning is indeed remelting, as the Reviewer suspected. In other words, when a region within a layer is re-scanned, it is melted twice using the same laser scan strategy. Also, the re-scanning is conducted region by region (or area by area) and not laser track by laser track. In the revised manuscript, we clarify this point by adding more details about our processing strategy (in the Methods) and by replacing “re-scanning” with “remelting” throughout.

Also – what is the time delay between each scan? The extent of the retained heat in the material locally will depend, critically, on the scanning strategy used. I would like this discussed more fully either in the methods or in the supplementary materials, because as it stands the method lacks sufficient detail to be replicated by a third party.

Response: The Reviewer makes an excellent point. Incidentally, work is ongoing in our group to investigate the “size effects” of our laser scanning strategies, being them aimed at programmable recrystallization, or at controlling other microstructural features (notably texture) which we are interested in. This work will be published in another paper.

To ensure reproducibility of the results reported in the current manuscript, we have revised our Method section to include more information on this point, which we report hereafter:

“In these cases, the completion of one entire layer (i.e., after scanning all samples) varied from 1 to 3 minutes, which was long enough for the scanned areas to cool down without introducing significant thermal buildups.”

Following the Reviewer's request, we have also expanded the discussion on this point in the first paragraph of page 7:

"In fact, we produced the two samples shown in Fig 1d and 1e using different LPBF machines. We highlight, however, that the exact laser parameters employed for strategy H and L differ when using different machines (as we detail in Methods) and should be revised depending on the size and geometry of the build. The smaller the laser-scanned areas to be consolidated, the higher the retained heat during processing, which may affect the solidification microstructure significantly."

Other than this minor quibble, the paper is very interesting and would find a ready audience. Its exploration of GNDs and recrystallisation to manipulate microstructure is a welcome contribution.

Response: We thank the Reviewer again for the positive and constructive feedback. We hope we were able to address their comments and make the manuscript suitable for publication.

Reviewers' Comments:

Reviewer #1:

Remarks to the Author:

The authors submitted a revised version of the submission 'Additive manufacturing of alloys with programmable microstructure and properties'. Further explanations and additional results are certainly appreciated. Nevertheless, there are several shortcomings that, from the reviewer's perspective' were not addressed.

1. The authors put effort in highlighting the novelty of the work. As mentioned before, there are novel aspects. The authors state that 'the ability to trigger recrystallization without having to mechanically deform the microstructure is major novelty of our work'. The reviewer disagrees in this regard. There are several studies that report on the recrystallization of additively manufactured materials without prior deformation, e.g. [<https://doi.org/10.1016/j.msea.2016.03.036>, <https://doi.org/10.1016/j.jallcom.2019.06.305>, <https://doi.org/10.1016/j.matchar.2021.110969>] and further studies can be found. Also, from a metal-physical perspective, it is easy to understand that varying LPBF conditions result in different dislocation densities and this has been reported frequently. Thus, there is often a sufficient driving force for static recrystallization. I believe that the authors overestimate the novelty of their work in this aspect.
2. The addition of the FEM approach is appreciated but not very useful in the context of the study, as it 'just' reproduces the trend observed from the experiments. An interesting contribution would be an approach that predicts the local dislocation densities, which currently does not exist. However, the recrystallization behaviour is determined by the local nucleation and growth conditions. These cannot be delivered by the applied FEM simulation. Furthermore, although there is a similar trend in the FEM output and EBSD results, the FEM model does not allow for simulating the correct dislocation density. In addition to the repeated heating-cooling cycles and volume change during phase transformation, local growth and coalescence of dendrites and grains during solidification cannot be considered, but is essential for the local distribution of dislocations, which, in turn, is important for the RX controlled microstructure design. Considering points 1 and 2, the reviewer is very sceptical when it comes to terms such as 'major novelty' and 'breakthrough' to describe the presented study.
3. As previously mentioned, the viscoplastic model is not necessary for the presented study, from the reviewer's perspective. However, it does not weaken the manuscript but distracts from the main aspect of the work. If I understand the authors correctly, the viscoplastic model was calibrated based on the experimental results and the interplay of layers with different properties is known from the field of heterostructured materials. It would add value to the study if the simulation was used before doing experiments to identify 'ideal' layered structures or if it is used to better understand the fundamental plastic deformation of these structures. However, that does not seem to be the case.

Reviewer #2:

Remarks to the Author:

The Authors have fully addressed the comments and suggestions made in my previous review and I am happy with how these changes have been made and incorporated into the paper. This is excellent and very interesting work.

Reviewers' comments

Authors' responses

Reviewer #1

The authors submitted a revised version of the submission 'Additive manufacturing of alloys with programmable microstructure and properties'. Further explanations and additional results are certainly appreciated. Nevertheless, there are several shortcomings that, from the reviewer's perspective' were not addressed.

Response: We thank the Reviewer for their appreciation of our revisions.

1. The authors put effort in highlighting the novelty of the work. As mentioned before, there are novel aspects. The authors state that 'the ability to trigger recrystallization without having to mechanically deform the microstructure is major novelty of our work'. The reviewer disagrees in this regard. There are several studies that report on the recrystallization of additively manufactured materials without prior deformation, e.g. [<https://doi.org/10.1016/j.msea.2016.03.036>,<https://doi.org/10.1016/j.jallcom.2019.06.305>, <https://doi.org/10.1016/j.matchar.2021.110969>] and further studies can be found. Also, from a metal-physical perspective, it is easy to understand that varying LPBF conditions result in different dislocation densities and this has been reported frequently. Thus, there is often a sufficient driving force for static recrystallization. I believe that the authors overestimate the novelty of their work in this aspect.

Response: For starters, we would like to point out that the statement highlighted by the Reviewer does not appear in the revised manuscript. It was in our response to one of the comments made in the first review round. Thus, we made no changes to the manuscript to tone down the related novelty claims.

In general, however, we understand the reason why the Reviewer disagrees with our statement and admit that we were not sufficiently precise. The statement should be read "the ability to trigger recrystallization site-specifically and at different temperatures without relying on mechanical strains is a major novelty of our work". We completely agree that the recrystallization can be achieved in additively manufactured materials without prior deformation, as reported in the example papers shared by the Reviewer. As long as the heat treatment temperature is high enough or the duration is long enough, recrystallization may be triggered even in as-cast materials that contain lower dislocation densities. In AM stainless steel 316L, despite of higher dislocation densities, recrystallization is generally triggered at a temperature over 1200 °C without prior deformation (e.g., <https://doi.org/10.1016/j.actamat.2020.11.018> and <https://doi.org/10.1016/j.msea.2017.10.002>). In our work, however, we showcase the ability to trigger recrystallization employing a temperature as low as 1050 °C and site-specifically (a feature that has not been shown so far). This is by virtue of the level of control we achieve over the dislocation density in the microstructure and the thermal stability of the alloy during LPBF.

On that note, the Reviewer indicates that varying LPBF parameters to generate different dislocation densities is quite intuitive. However, within a typical processing window in which laser scan speed and power are varied to achieve high-density builds, dislocation densities only differ slightly (as shown, for instance, in our previous work <https://doi.org/10.1016/j.actamat.2020.09.015>). In this work, we can vary the as-built dislocation density by a factor ~ 2 , which denotes a rarely reported level of control.

To make our statement of novelties more specific, in the revised manuscript we now state:

On page 7 of the revised manuscript, “*By contrast, our LPBF strategies may be alternated in space at relatively high spatial resolution (as shown in Fig. 1e) and offer an entirely deformation-free, site-specific control over the microstructure, which has not been achieved so far.*”

2. The addition of the FEM approach is appreciated but not very useful in the context of the study, as it ‘just’ reproduces the trend observed from the experiments. An interesting contribution would be an approach that predicts the local dislocation densities, which currently does not exist. However, the recrystallization behaviour is determined by the local nucleation and growth conditions. These cannot be delivered by the applied FEM simulation. Furthermore, although there is a similar trend in the FEM output and EBSD results, the FEM model does not allow for simulating the correct dislocation density. In addition to the repeated heating-cooling cycles and volume change during phase transformation, local growth and coalescence of dendrites and grains during solidification cannot be considered, but is essential for the local distribution of dislocations, which, in turn, is important for the RX controlled microstructure design. Considering points 1 and 2, the reviewer is very sceptical when it comes to terms such as ‘major novelty’ and ‘breakthrough’ to describe the presented study.

Response: First, we would like to point out that we added the FEM work in response to the Reviewer’s comment in the previous revision round, when they stated: “*There is no attempt to further describe the evolution of the dislocation density in a physics-based manner and neither a related modelling approach, which would be a major contribution to the field of research*”. We agreed with that comment and developed the FEM work to do exactly that. The model predicts the evolution of the dislocation density in the material as a function of the process parameters we employed and on the basis of physics-based model.

Secondly, we would argue that being able to model the conditions for nucleation of grains during recrystallization by capturing the `_local_` dislocation density (e.g., within a single melt pool) as a function of the solidification kinetics and of the `_local_` solidification microstructure is a herculean endeavor and—albeit worthwhile pursuing in the future—one that is beyond the scope of this work.

We also highlight that we did discuss the possible reason for the discrepancies between the simulation predictions and our dislocation density measurements on page 8 of the revised manuscript: “*the FEM model does not capture the non-uniform*

distribution of plastic strain in each melt pool.” and “crystal misorientation (and thus GND density) is highest at melt pool centerlines.” These considerations are aligned with what the Reviewer is saying, namely that our model does not capture the `_local_` dislocation density evolution as a function of the solidification microstructure. Thus, we are positive that we are not overselling the contributions of our work.

3. As previously mentioned, the viscoplastic model is not necessary for the presented study, from the reviewer’s perspective. However, it does not weaken the manuscript but distracts from the main aspect of the work. If I understand the authors correctly, the viscoplastic model was calibrated based on the experimental results and the interplay of layers with different properties is known from the field of heterostructured materials. It would add value to the study if the simulation was used before doing experiments to identify ‘ideal’ layered structures or if it is used to better understand the fundamental plastic deformation of these structures. However, that does not seem to be the case.

Response: We appreciate the Reviewer’s comment and understand their perspective. The Reviewer is correct that the viscoplasticity model is calibrated on our experimental data. While it is not currently capable to predict the behavior of layered microstructures without experimental data or to further explain the fundamental plastic deformation, we believe that the model adds value to the manuscript. First, it proves that the extra work hardening stems from the microstructure-interfaces. Second, it indeed can be used to predict the performance improvement in different layered architectures (as shown in Fig. 5f). Third, it pinpoints the shortcoming stemming from the limited spatial resolution of the LPBF machines we used. In view of these points, we decide to keep the model in the manuscript.

Reviewer #2

The Authors have fully addressed the comments and suggestions made in my previous review and I am happy with how these changes have been made and incorporated into the paper. This is excellent and very interesting work.

Response: We thank the Reviewer for the very strong endorsement on publishing our work.